# FOURIER PINNS: FROM STRONG BOUNDARY CONDITIONS TO ADAPTIVE FOURIER BASES

## ABSTRACT

Interest in Physics-Informed Neural Networks (PINNs) is rising as a mesh-free alternative to traditional numerical solvers for partial differential equations (PDEs). While successful, PINNs often struggle to learn high-frequency and multi-scale target solutions—which, according to prior analysis, might arise from competition during optimization between the weakly enforced boundary loss and residual loss terms. By creatively modifying the neural network architecture, some simple boundary conditions (BCs) can be satisfied exactly without jointly optimizing an additional loss term, thus avoiding the aforementioned competition. Motivated by this analysis, we first study a strong BC version of PINNs for Dirichlet BCs and observe a consistent improvement compared to the standard PINNs. We conducted a Fourier analysis and found that strong BC PINNs can better learn the amplitudes of high-frequency components of the target solutions. While BC PINNs provide improvement, constructing such architectures is an intricate process made difficult (if not impossible) by certain BCs and domain geometries. Enlightened by our analysis, we propose Fourier PINNs — a simple, general, yet powerful method that augments PINNs with pre-specified, dense Fourier bases. Our proposed architecture likewise better learns high-frequency components but places no restrictions on the particular BCs. We developed an adaptive learning and basis selection algorithm based on alternating NN basis optimization, Fourier and NN basis coefficient estimation, and coefficient truncation. This scheme can flexibly identify the significant frequencies while weakening the nominal to better capture the target solution's power spectrum. We show the advantage of our approach in a set of systematic experiments.

## 1 INTRODUCTION

Physics-informed neural networks (PINNs) (Raissi et al., 2019a) are emergent mesh-free approaches to solving partial differential equations (PDE)s. They have shown successful in many scientific and engineering problems, such as bio-engineering (Sahli Costabal et al., 2020; Kissas et al., 2020), fluids mechanics (Raissi et al., 2019b; Sun et al., 2020; Raissi et al., 2020), fractional PDEs (Pang et al., 2019b; 2020), and material design (Fang & Zhan, 2019; Liu & Wang, 2019). The PINN framework uses neural networks (NNs) to estimate PDE solutions, in light of the universal approximation ability of the NNs. Specifically, consider a PDE of the following general form,

$$\mathcal{F}[u](\mathbf{x}) = f(\mathbf{x}) \ (\mathbf{x} \in \Omega), \quad u(\mathbf{x}) = g(\mathbf{x}) \ (\mathbf{x} \in \partial\Omega) \tag{1}$$

where $\mathcal{F}$ is the differential operator for the PDE, $\Omega$ is the domain, $\partial\Omega$ is the boundary of the domain. To solve the PDE, the PINN uses a deep neural network $\widehat{u}_{\boldsymbol{\theta}}(\mathbf{x})$ to represent the solution $u$, samples $N$ collocation points $\{\mathbf{x}_c^i\}_{i=1}^N$ from $\Omega$ and $M$ points $\{\mathbf{x}_b^i\}_{i=1}^M$ from $\partial\Omega$, and minimizes the loss,

$$\boldsymbol{\theta}^* = \underset{\boldsymbol{\theta}}{\operatorname{argmin}} \ L_b(\boldsymbol{\theta}) + L_r(\boldsymbol{\theta}) \tag{2}$$

where $L_b(\boldsymbol{\theta}) = \frac{1}{M}\sum_{j=1}^M \left(\widehat{u}_{\boldsymbol{\theta}}(\mathbf{x}_b^j) - g(\mathbf{x}_b^j)\right)^2$ is the boundary loss to fit the boundary condition, and $L_r(\boldsymbol{\theta}) = \frac{1}{N}\sum_{j=1}^N \left(\mathcal{F}[\widehat{u}_{\boldsymbol{\theta}}](\mathbf{x}_c^j) - f(\mathbf{x}_c^j)\right)^2$ is the residual loss to fit the equation.

Despite their success, the training of PINNs is often unstable, and the performance can be poor from time to time, especially when solutions includes high-frequency and multi-scale components.

From the optimization perspective, Wang et al. (2020a) pointed out that due to the imbalance of the gradient magnitudes of the boundary loss and residual loss (the latter is often much larger), the training can be dominated by the residual loss and hence underfits the boundary condition. Wang et al. (2020c) confirmed this conclusion from a neural tangent kernel (NTK) analysis on the training behaviors of PINNs with wide networks. They found the eigenvalues of the residual kernel matrix are often dominant, which can cause the training to mainly fit the residual loss. On the other hand, Rahaman et al. (2019) found the "spectrum bias" in learning standard NNs, namely, the low frequency information can be easily learned from data but grasping the high frequencies is much slower and harder. To alleviate this issue, Tancik et al. (2020) proposed to randomly sample a set of high frequencies from a large-variance Gaussian distribution, and use these frequencies to construct random Fourier features as the input to the subsequent NN. Wang et al. (2021) gave a justification of this approach via NTK analysis, and used multiple Gaussian variances to sample the frequencies and construct Fourier features so as to capture multi-scale information in the PINN framework. While effective, the performance of this method is sensitive to the number and scales of the Gaussian variances, which are often difficult to choose because the solution is unknown apriori.

The contributions of our work are as follows:

- Motivated by the prior analysis from the optimization perspective, we investigated a strong boundary condition (BC) version of PINNs for simple Dirichlet BC's. This PINN variant satisfies the boundary conditions exactly (through specific NN architecture construction) and does not require a boundary loss term during optimization—thus avoiding the competition with the residual loss term. We observed significant improvement upon the standard PINN, especially for higher frequency problems.
- Different from the previous investigation, we conducted a Fourier analysis on our strong BC PINNs using a specific boundary function—the function that constrains the NN output to satisfying the boundary conditions exactly. Through Fourier series and convolution theory, we found that, interestingly, multiplying the NN with the boundary function enables faster and more accurate learning of the coefficients of higher frequencies in the target solution. By contrast, standard PINNs exhibit hardship in capturing correct coefficients in the high frequency domain.
- Enlightened by our analysis, we developed Fourier PINNs—a simple, general, yet powerful extension of PINNs, independent of any particular boundary condition (unlike Strong BC PINNs). The solution is modeled as an NN plus the linear combination of a set of dense Fourier bases, where the frequencies are evenly sampled from a large range. We developed an adaptive learning and basis selection algorithm, which alternately optimize the NN basis parameters and the coefficients of the NN and Fourier bases, meanwhile pruning useless or insignificant bases. In this way, our method can quickly identify important frequencies, supplement frequencies missed by the NN, and improve the amplitude estimation. We only need to specify a large enough range and small enough spacing for the Fourier bases, without the need for worrying about the actual number and scales of the frequencies in the true solution as in previous methods. All these can be automatically inferred during training.
- We evaluated FourierPINNs in several benchmark PDEs with high-frequency and multi-frequency solutions and solutions that couple high-frequency components with plain functions. In all the cases, Fourier PINNs consistently achieve reasonable and good solution errors, *e.g.*, $\sim 10^{-3}$ or $\sim 10^{-2}$. As a comparison, the standard PINNs always failed, while the strong BC PINNs were much worse than our method (though better than the standard PINNs). The PINNs with random Fourier features often failed under a variety of choices of the Gaussian variance number and scales. The performance is highly sensitive to this choice. We also tested PINNs with large boundary loss weights (Wight & Zhao, 2020), and with an adaptive activation function (Jagtap et al., 2020). FourierPINNs consistently outperformed both methods.

## 2   Strong Boundary Condition PINNs

According to prior analysis (Wang et al., 2020a;c), the instability of PINNs is likely from the competition between the weakly-enforced boundary loss and the residual loss during optimization. To sidestep this issue, a natural idea is to design a surrogate model that satisfies the boundary condition, and hence the training no longer needs a weakly-forced boundary loss. To this end, we consider a

simple Dirichlet boundary condition for 1d problems,

$$x \in [a, b], \quad u(a) = u(b) = 0. \tag{3}$$

To solve the equation $\mathcal{F}[u](x) = f(x)$, we construct a surrogate model by

$$\widetilde{u}_{\boldsymbol{\theta}}(x) = B(x)\mathcal{N}_{\boldsymbol{\theta}}(x) \tag{4}$$

where $\mathcal{N}_{\boldsymbol{\theta}}(x)$ is a neural network parameterized by $\boldsymbol{\theta}$, and

$$B(x) = (x - a)(b - x) \tag{5}$$

is a boundary function. We can see that by construction $\widetilde{u}_{\boldsymbol{\theta}}(a) = \widetilde{u}_{\boldsymbol{\theta}}(b) = 0$, namely, the boundary condition (3) is automatically satisfied. To estimate the parameters of $\widetilde{u}$, we only need to minimize the residual loss $L_r(\boldsymbol{\theta})$ (see (2)). The construction is straightforward. Similar strategies has been used in the recent work (Lu et al., 2021b) to incorporate hard constraints for topology optimization and inverse design, and early works, *e.g.*, (Lagaris et al., 1998; 2000). However, here we want to investigate if such construction can benefit solving forward problems, especially when the solution includes high frequency information.

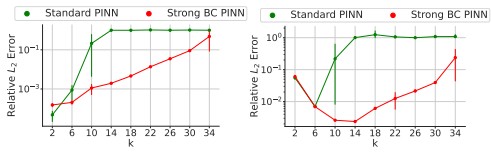

To this end, we examined the strong boundary condition (BC) PINN (4) on a 1d Poisson equation,

$$\Delta u = f(x), \ \ x \in [0, 2\pi], \tag{6}$$

and a 1d Allen-Cahn equation,

$$u_{xx} + u(u^2 - 1) = f(x), \ \ x \in [0, 2\pi], \tag{7}$$

(a) 1D Poisson   (b) 1D Allen-Cahn

Figure 1: Solution error with frequency $k$.

both subject to the Dirichlet condition (3) where $a = 0$ and $b = 2\pi$. We consider the fabricated solution in the form of $u(x) = \sin(kx)$. Therefore, by varying $k$, we can examine the performance the strong BC PINN when the solution includes different frequency information. We compared against the standard PINN, which explicitly incorporates the weakly-enforced boundary loss and residual loss. We used the same NN architecture for the strong BC PINN and the standard PINN, which includes two hidden layers, 100 neurons per layer, and `tanh` activation. Both methods used the same set of 10K collocation points randomly sampled from the domain. We varied $k$ from 2 to 34, and ran each method for five times. The average relative $L_2$ error and the standard deviation is reported in Fig. 1. As we can see, the strong BC PINN consistently outperforms the standard PINN, including bigger choices of $k$. That means, the strong BC PINN can more accurately capture the high frequency information in the solution.

## 3 Fourier Analysis of Strong BC PINNs

Now, we attempt to explain why the strong BC PINNs can improve upon the standard PINNs. We look into this problem via Fourier analysis. A plain NN model is known to easily capture the low frequency information yet difficult/very slow to grasp the high frequency information (Rahaman et al., 2019). We first investigate if physics-informed NNs exhibit similar issues. Fig. 2a shows a typical example of the frequency spectrum (obtained from discrete Fourier transform), which comes from the standard PINN trained to solve the 1D Poisson in Sec. 2 with true solution $u(x) = \sin(15x)$. We can see that although the PINN is able to identify the target frequency, its coefficients for higher frequencies are too big, *i.e.*, heavy tails. This leads to the inferior solution accuracy. By contrast, from the frequency spectrum of the strong BC PINN (Fig. 2a), we can see that not only does the strong BC PINN captures the target frequency (as accurate as the standard PINN), its coefficients for higher frequencies decay much faster, and hence the influence of unnecessary high frequencies are greatly weakened (or even excluded), leading to much better accuracy.

To understand why the boundary function (5) (after multiplied to the NN in (4)) can help obtain better coefficients for high frequencies, we first seek to represent the boundary function (5) with an infinite Fourier series (synthesis analysis),

$$B(x) = B_{\infty}(x) = \sum_{n=-\infty}^{+\infty} \hat{B}[n] \cdot e^{inx}$$

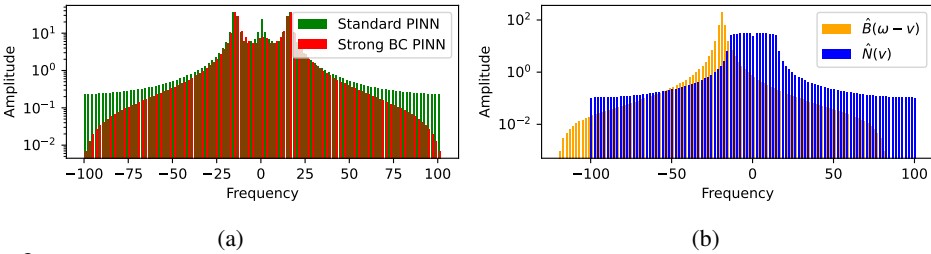

(a)              (b)

Figure 2: The frequency spectrum of the learned solutoin surrogate (a) and the convolution operation in the Strong BC PINN with $\omega = 15$. In both cases, the ground-truth solution is $\sin(15x)$.

where $i$ indicates the imaginary part, and we have

$$\hat{B}[n] = \frac{1}{2\pi} \int_{2\pi} x(2\pi - x) \cdot e^{-inx} \mathrm{d}x = -\frac{2}{n^2}.$$

Here we expand the boundary function $B(x)$ to be periodic (*i.e.*, duplicating its definition in $[0, 2\pi]$ to other intervals) and the period is $2\pi$. We can now obtain the Fourier transform of $B_\infty(x)$,

$$\widehat{B}(\omega) = \sum_{n=-\infty}^{+\infty} \hat{B}[n] \cdot \delta\left(\omega - \frac{n}{2\pi}\right) \tag{8}$$

where $\delta(\cdot)$ is the Dirac delta function. Since $|\hat{B}[n]| \propto 1/n^2$, we can see that the amplitude of the frequencies of $B(x)$ decays quadratically fast with the increase of the absolute frequency value.

We then look at the frequency spectrum of the surrogate model $\widetilde{u}(x)$ in (4). Since it is the product of $B(x)$ and $\mathcal{N}(x)$, we can use the convolution theorem (McGillem & Cooper, 1991) to obtain

$$\widehat{\widetilde{u}}(\omega) = \widehat{B} * \widehat{\mathcal{N}} = \int_{-\infty}^{\infty} \widehat{B}(\omega - \nu)\widehat{\mathcal{N}}(\nu)\mathrm{d}\nu \tag{9}$$

where $*$ means convolution, and $\widehat{\mathcal{N}}$ is the Fourier transform of the neural network. Suppose $\omega > 0$, we know that $\widehat{B}(w - \nu)$ is obtained by first reflecting $\widehat{B}(\nu)$ about the y-axis, and then move it $\omega$ to the left. Since $\widehat{B}$ is symmetric (see (8)), we actually just move the original function to the left. Then, we integrate the whole $\widehat{\mathcal{N}}(\nu)$ weighted by $\widehat{B}(w - \nu)$. The larger the frequency $\omega$, the more left $\widehat{B}$ is moved to so as to obtain $\widehat{B}(w - \nu)$. That means, the more tail (right) part of the $\widehat{B}$ is used to weight $\widehat{\mathcal{N}}(\nu)$ during the integration. Since the frequency coefficients of $\widehat{B}$ decays quadratically fast ($|\hat{B}[n]| \propto 1/n^2$), the larger the portion of the tails used, the smaller the integration result. That means, with the help of the boundary function $B$, the corresponding amplitude $\widehat{\widetilde{u}}(\omega)$ decrease fast when $\omega$ increase. See Fig. 2b for an illustration. The same conclusion applies when we consider $\omega < 0$ and $|\omega|$ increase. In this way, during the course of training, the boundary function pushes the surrogate model (4) to inhibit/weaken irrelevant high frequency components so as to reduce flat tails and to better capture the frequency spectrum.

## 4   Fourier PINNs

Based on our analysis (Sec. 3), the bottleneck of the standard PINN model is the inefficiency of learning the high-frequency components of the solution — the estimated amplitudes are overly big. While the strong BC PINNs alleviate this issue by multiplying the NN with a boundary function to enable a faster decay of the amplitudes, the benefit can still be restricted since the decay rate is up to the boundary function itself rather than is driven by the ground-truth solution. As can be seen from Fig. 1, as $k$ grows, the performance of the strong BC PINN degraded, implying that it is not powerful enough. In addition, the strong BC PINNs are restricted to particular boundary conditions, *e.g.*, the Dirichlet conditions (3), which severely limit its applicability. To overcome these problems, we are enlightened by our analysis and propose Fourier PINNs, a simple, general yet powerful method, which can handle various boundary conditions like the standard PINNs. Specifically, in order to flexibly and comprehensively capture the frequency spectrum, we first introduce a set of dense frequency candidates, $\{\omega_n\}_{n=1}^Q$, which are evenly sampled from a large range $[-K, K]$ with spacing $\Delta$. We then augment the NN with a set of Fourier bases,

$$\widetilde{u}(\mathbf{x}) = \mathcal{N}_{\boldsymbol{\theta}}(\mathbf{x}) + \sum_{n=1}^Q a_j \cos(2\pi\omega_n x) + b_j \sin(2\pi\omega_n x). \tag{10}$$

Since the NN can be viewed as the linear combination of a set of nonlinear bases, the surrogate can be further written as

$$\widetilde{u}(\mathbf{x}) = \sum\nolimits_{j=1}^{H+2Q} w_j \phi_j(\mathbf{x}) \tag{11}$$

where for $1 \leq j \leq H$, $\phi_j$ is an NN basis, namely, the $j$-th output of the second last layer, for $H + 1 \leq j \leq H + Q$, $\phi_j(x) = \cos(2\pi\omega_n x)$ $(n = j - H)$, for $H + Q + 1 \leq j \leq H + 2Q$, $\phi_j(x) = \sin(2\pi\omega_n x)$ $(n = j - H - Q)$, and $w_j$ is the coefficient of each basis. We then use a loss similar to PINNs (see (2)) to estimate the parameters,

$$L(\boldsymbol{\psi}, \mathbf{w}) = L_b(\boldsymbol{\psi}, \mathbf{w}) + L_r(\boldsymbol{\psi}, \mathbf{w}) + \frac{\alpha}{2}\|\mathbf{w}\|^2$$

$$= \frac{1}{M}\sum\nolimits_{i=1}^{M} \left(\widetilde{u}(\mathbf{x}_b^i) - g(\mathbf{x}_b^i)\right)^2 + \frac{1}{N}\sum\nolimits_{i=1}^{N} \left(\mathcal{F}[\widetilde{u}](\mathbf{x}_c^i) - f(\mathbf{x}_c^i)\right)^2 + \frac{\alpha}{2}\|\mathbf{w}\|^2, \tag{12}$$

where $\boldsymbol{\psi}$ denotes the parameters of the NN bases, $\mathbf{w} = (w_1, \ldots, w_{H+2Q})^\top$ all the basis coefficients, $\|\mathbf{w}\|^2$ is the square norm to regularize $\mathbf{w}$, and $\alpha$ is the regularization strength.

The ground-truth frequencies can be much fewer than the candidates. From our analysis, to improve PINNs, it is critical to prune or inhibit useless frequencies, *e.g.*, those that are overly big (see Fig. 2a). In order to flexibly identify the significant and prune or weaken the insignificant frequencies, we develop an adaptive learning and basis selection algorithm. Specifically, we jointly optimize $\boldsymbol{\psi}$ and $\mathbf{w}$ to a stage. Then we fix all the bases (*i.e.*, with $\boldsymbol{\psi}$ fixed), and iteratively truncate the basis coefficients $\mathbf{w}$ with a threshold (*e.g.*, $|w_j| \leq 10^{-3}$), and re-update the remaining coefficients. We repeat this procedure until a maximum number of iterations are finished.

When the PDE operator $\mathcal{F}$ is linear, the updates of $\mathbf{w}$ can be simply done via least square ridge regression, because according to (11), $\mathcal{F}[\widetilde{u}](\cdot) = \sum_j w_j \mathcal{F}[\phi_j](\cdot)$ is linear in $\mathbf{w}$, and the loss has the form $L(\boldsymbol{\psi}, \mathbf{w}) = \frac{\alpha}{2}\|\mathbf{w}\|^2 + \frac{1}{M}\sum_{i=1}^{M} \left(\sum_j w_j \cdot \phi_j(\mathbf{x}_b^i) - g(\mathbf{x}_b^i)\right)^2 + \frac{1}{N}\sum_{i=1}^{N} \left(\sum_j w_j \cdot \mathcal{F}[\phi_j](\mathbf{x}_c^i) - f(\mathbf{x}_c^i)\right)^2$. Note that the method of alternating least square and truncation has achieved a great success in sparse linear regression for data-drive operator selection and governing equation discovery (Brunton et al., 2016; Chen et al., 2021). We use a similar strategy rather than the conventional $L_1$ regularization for simplicity and efficiency. When the operator $\mathcal{F}$ is nonlinear yet can be decomposed as

$$\mathcal{F}[\cdot] = \mathcal{G}[\cdot] + \mathcal{S}[\cdot] \tag{13}$$

where $\mathcal{G}$ is a linear operator and $\mathcal{S}$ is the nonlinear part, *e.g.*, in the Allen-Cahn equation (7), $\mathcal{G}[u] = u_{xx}$ and $\mathcal{S}[u] = u(u^2 - 1)$, we can still conduct efficient least square ridge regression to update $\mathbf{w}$. Specifically, we evaluate $\mathcal{S}[\widetilde{u}]$ with current $\mathbf{w}$ on the collocation points, fix the results, and combine with the source function values on the collocation points to construct the regression output in the residual loss,

$$L_r = \frac{1}{N}\sum\nolimits_{i=1}^{N} \left(\sum_j w_j \cdot \mathcal{G}[\phi_j](\mathbf{x}_c^i) - \hat{f}_i\right)^2 \tag{14}$$

where $\hat{f}_i = -\mathcal{S}[\widetilde{u}](\mathbf{x}_c^i) + f(\mathbf{x}_c^i)$. Here, since $\hat{f}_i$ is determined from the current $\mathbf{w}$ but fixed as a constant in the least square estimation, the update of $\mathbf{w}$ is in essence a type of fixed point iteration. In our experiments, we found this method is efficient and robust. When $\mathcal{F}$ does not include any linear operator (which might be rare), we can use any continues optimization algorithm, *e.g.*, L-BFGS, to update $\mathbf{w}$. Finally, our adaptive learning and basis selection is summarized in Algorithm 1.

While we present our method in the single input case, it is straightforward to extend to multiple inputs. Consider $\mathbf{x} = [x_1; x_2]$ for an example. We use the same set of frequencies to construct Fourier bases for each single input, $\boldsymbol{\phi}(x_1) = [\cos(2\pi w_1 x_1), \sin(2\pi w_1 x_1), \ldots, \cos(2\pi w_Q x_1), \sin(2\pi w_Q x_1)]$, and $\boldsymbol{\phi}(x_2) = [\cos(2\pi w_1 x_2), \sin(2\pi w_1 x_2), \ldots, \cos(2\pi w_Q x_2), \sin(2\pi w_Q x_2)]$. Then we apply a cross-product to obtain the Fourier bases for $\mathbf{x}$. The surrogate model is given by

$$\widetilde{u}(\mathbf{x}) = \mathcal{N}_{\boldsymbol{\theta}}(\mathbf{x}) + \boldsymbol{\beta}^\top \text{vec}\left(\boldsymbol{\phi}(x_1)\boldsymbol{\phi}(x_2)^\top\right) \tag{15}$$

where $\text{vec}(\cdot)$ is the vectorization, and $\boldsymbol{\beta}$ are the coefficients of the Fourier bases. Note that the number of bases grows fast with the increase of input dimension, and can be too large for high input dimensions. In that case, we can pre-prune the bases according to the total frequency or re-organize the coefficients $\boldsymbol{\beta}$ into a tensor or matrix, and introduce a low-rank decomposition structure to parameterize $\boldsymbol{\beta}$, so as to save the cost (Novikov et al., 2015; Ye et al., 2018).

---

**Algorithm 1** FourierPINN($\delta$, $T$, $A$, $E$)

---

Initialize $\boldsymbol{\psi}$ and $\mathbf{w}$ randomly.
Run ADAM for $T$ epochs to jointly optimize $\boldsymbol{\psi}$ and $\mathbf{w}$.
**repeat**
    **for** $A$ iterations **do**
        Fix $\boldsymbol{\psi}$, estimate $\mathbf{w}$ with least square ridge regression.
        For each $w_j$, if $|w_j| < \delta$, prune the corresponding basis $\phi_j(\cdot)$ and delete $w_j$ from $\mathbf{w}$.
    **end for**
    Run ADAM for $T$ epochs to jointly optimize $\boldsymbol{\psi}$ and $\mathbf{w}$.
**until** $E$ iterations are done
Run L-BFGS until convergence

---

## 5 Related Work

There is a large body of related work. Due to the space limit, we briefly discuss the most related lines of work. PINNs have shown promise as a computationally efficient alternative to expensive mesh-based numerical methods such as finite elements and finite volumes (Reddy, 2019). PINNs (typically) work by *softly* constraining neural networks to abide by physical constraints given well-studied governing equations — a strategy that is beneficial particularly in the small data regime (Rackauckas et al., 2020; Lu et al., 2021a; Weinan et al., 2017; Raissi et al., 2019a; Hennigh et al., 2021). Although PINNs have many success stories, *e.g.*, (Raissi et al., 2020; Chen et al., 2020; Jin et al., 2021; Sirignano & Spiliopoulos, 2018; Zhu et al., 2019; Geneva & Zabaras, 2020; Sahli Costabal et al., 2020), the training of the PINNs is known to be challenging, which is partly due to that applying differential operators over the NN in the residual can complicate the loss landscape (Krishnapriyan et al., 2021). Recent works have analyzed common failure modes of PINNs which include modeling problems exhibiting high-frequency, multi-scale, chaotic, or turbulent (likely among many other) behaviours (Wang et al., 2020c;b;a; 2022), or when the governing PDEs are stiff (Krishnapriyan et al., 2021; Mojgani et al., 2022).

One class of approaches to mitigate the training challenge of PINNs is to set different weights for the boundary and residual loss terms. For example, Wight & Zhao (2020) suggested to set a large weight for the boundary loss to prevent the dominance of the residual loss. Wang et al. (2020a) proposed a dynamic weighting scheme based on the gradient statistics of the loss terms. Wang et al. (2020c) developed an adaptive weighting approach based on the eigen-values of NTK. Liu & Wang (2021) employed a mini-max optimization and update the loss weights via stochastic ascent. McClenny & Braga-Neto (2020) used a multiplicative soft attention mask to dynamically re-weight the loss term on each data point and collocation point. Another strategy is to modify the NN architecture so as to exactly satisfy the boundary conditions (Lu et al., 2021b; Lyu et al., 2020; Lagaris et al., 1998; 2000; McFall & Mahan, 2009; Berg & Nyström, 2018; Lagari et al., 2020; Mojgani et al., 2022; Lagaris et al., 1997; Pang et al., 2019a; Yu et al., 2022). In this way, the boundary loss term is no longer needed in training and so its competition with the residual loss is eliminated. In our work, we investigated such a version for PINNs in learning high-frequency and multi-scale solutions. However, these methods are restricted to particular types of boundary conditions, *e.g.*, the Dirichlet conditions, hence are less flexible than the original PINN framework. Tancik et al. (2020); Wang et al. (2021) used Gaussian distributions to construct random Fourier features to improve the learning of the high-frequency and multi-scale information. The number of Gaussian variances and their scales are critical to the success of these methods. But the hyperparameters can be quite difficult to choose.

## 6 Experiment

To evaluate our method, we benchmarked against state-of-the-art PINN models for solving high-frequency and/or multi-scale solutions: (1) standard PINNs, (2) Random Fourier Feature PINNs (RFF-PINN) (Wang et al., 2021). To ensure RFF-PINN achieves

| Number | Scales |
|--------|--------|
| 1 | $1, 20, 50, 100, rand(1, [1, K])$ |
| 2 | $rand(3, \{1, 20, 50, 100, rand(1, [1, K])\}), rand(2, [1, K])$ |
| 3 | $rand(3, \{1, 20, 50, 100, rand(1, [1, K])\}), rand(2, [1, K])$ |
| 5 | $rand(2, [1, K]) \times \{1, 20, 50, 100\}, rand(5, [1, K])$ |

Table 1: The number and scales of the Gaussian variances used in RFF-PINN, where $rand(k, \mathcal{A})$ means randomly selecting $k$ elements from the set $\mathcal{A}$ without replacement, $\times$ denotes the Cartesian product, and $K$ is the maximum candidate frequency used by FourierPINN.

the best performance, we followed (Wang et al., 2020c) to dynamically re-weight the loss terms

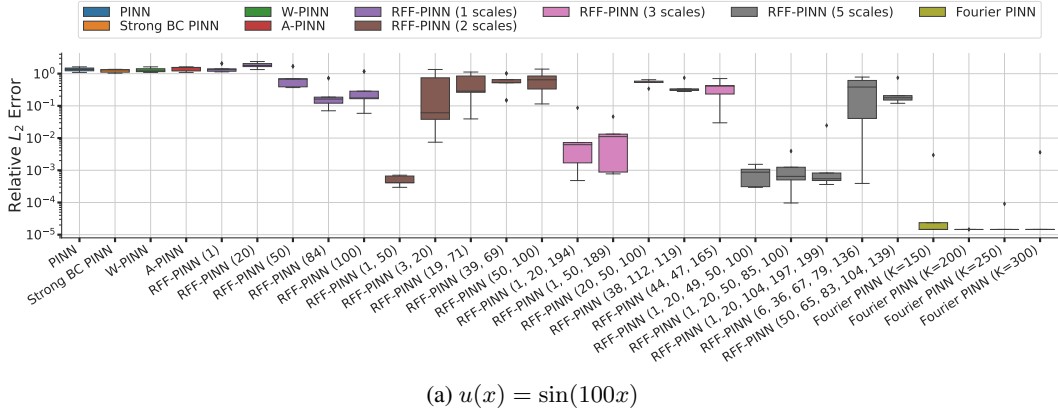

(a) $u(x) = \sin(100x)$

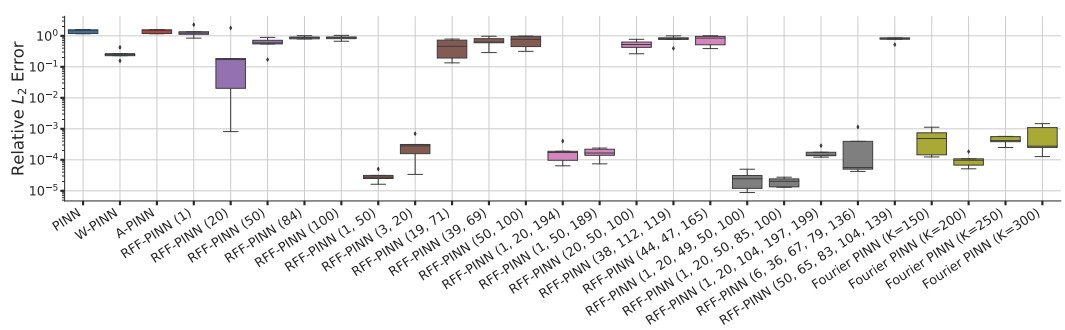

(b) $u(x) = \sin(x) + 0.1\sin(20x) + 0.05\cos(100x)$

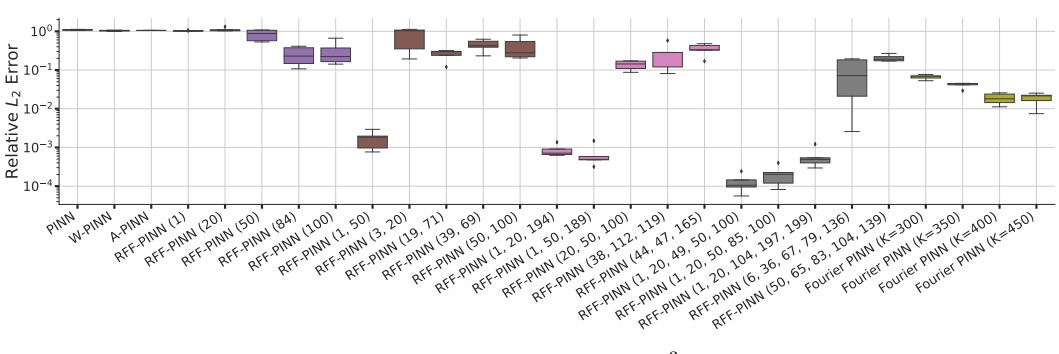

(c) $u(x) = 5x\cos(100x) + x^3$

Figure 3: Solving 1D Poisson equation.

based on NTK eigen-values (Wang et al., 2020c), as suggested by the authors. (3) Weighted PINNs (W-PINN) that down-weight the residual loss to reduce its dominance and to better fit the boundary loss. (4) Adaptive PINNs (A-PINN) with parameterized activation functions to increase the NN capacity and to be less prone to gradient vanishing and exploding. All the methods were implemented by PyTorch. For W-PINN, we varied the weight of the residual loss from $\{10^{-1}, 10^{-3}, 10^{-4}\}$. For A-PINN, we introduced a learned parameter for the activation function in each layer and update these parameters jointly during training. To run RFF-PINNs, we need to specify the number and scales of the Gaussian variances to construct the random features. To test a broad coverage, we varied the number of variances from $\{1, 2, 3, 5\}$. For each number, we set the variances to be the commonly used values suggested by authors, combined with randomly sampled ones. The detailed specification is given by Table 1. There are in total 20 settings.

## 6.1 1D Equations

We first tested with a set of 1D Poisson equations (see (6)). We varied the form of the true solution in the following, and examined the accuracy of each method.

$$u(x) = \sin(100x) \quad \text{(single scale)} \tag{16}$$
$$u(x) = \sin(x) + 0.1\sin(20x) + 0.05\cos(100x) \quad \text{(multi-scale)} \tag{17}$$
$$u(x) = 5x\cos(100x) + x^3 \quad \text{(hybrid)} \tag{18}$$

The boundary conditions are derived from the true solution. Note for the single scale case (16), the boundary condition is still Dirichlet, and hence we compared with the strong BC PINN. For all the methods, we used a neural network with two hidden layers, with 100 neurons per layer, and tanh activation function. For PINN, W-PINN, A-PINN and RFF-PINN, we first ran ADAM optimization for 40K epochs (learning rate $10^{-3}$), and then L-BFGS until convergence (the tolerance level $10^{-9}$). For FourierPINN, we set $\alpha = 0.1$, the number of iterations $T$ to 1K, the inner iteration number $A$ to 5, the truncation threshold $\delta$ to $10^{-4}$, and the outer iteration number $E$ to 40. For RFF-PINN, we set $K = 150$ for (16) and (17), and $K = 300$ for (18); see Table 1. We used the same $K$ for FourierPINN to specify the range of the frequency candidates $[-K, K]$. To further examine the stability, we also tested $K = 200, 250, 300$ for (16) and (17), and $K = 350, 400, 450$ for (18). The spacing is 1. We ran every method for five times, and each time with a different random initialization. We report the results in box plots in Fig. 3. We also examined 1D Allen-Cahn equation with $u(x) = \sin(100x)$. The result is reported in Fig. 5 of the Appendix. Note that, for W-PINN, we only report the setting that gives the best performance to save space.

We can see that, first, standard PINN, W-PINN, A-PINN and strong BC PINN all failed to obtain good solutions, and the relative error is large for all the cases (*e.g.*, $\sim 1.0$). It implies that, when the true solution includes very high frequency signals, these methods are not capable enough. Second, while with some number and scale choices, RFF-PINN can achieve very small relative errors, in most choices, which take 60% - 70%, RFF-PINN failed, and the solution error is large. The results show that the success of RFF-PINN is very sensitive to the choice of number and scales, and there seems no patterns. For example, for the single-scale true solution (see Fig. 3a), the successful cases of RFF-PINN include two, three or even five scale settings, *e.g.*, (1, 20, 194), yet it failed with all the single-scale settings. This can bring difficulties in practice. As a comparison, FourierPINN always achieves a reasonable, good solution accuracy in all the cases: $\sim 10^{-4}$ for the single-scale case, $\sim 10^{-3}$ for the multi-scale case, and $\sim 10^{-2}$ for the hybrid case. Our method is also robust to the choice of the frequency range $K$. As long as $K$ is large enough, our method nearly always selected the target frequency, and returned similar solution accuracy. Note that, to focus on the evaluation of the key idea, *FourierPINN did NOT employ any re-weighting scheme to further improve the accuracy, such as the NTK re-weighting used in RFF-PINN or mini-max updates.* But it is straightforward to integrate them into the proposed algorithmic framework.

## 6.2 2D Equations

We next tested on 2D Poisson and Allen-Cahn equations, which are give by

$$\Delta u = f(x, y), \quad x, y \in [0, 2\pi], \tag{19}$$
$$u_{xx} + u_{yy} + u(u^2 - 1) = f(x, y), \quad x, y \in [0, 2\pi], \tag{20}$$

respectively. We tested with the following ground-truth solutions,

$$u(x, y) = \sin(100x)\sin(100y), \quad \text{(2D Poisson)} \tag{21}$$
$$u(x, y) = (\sin(x) + 0.1\sin(20x) + \cos(100x))$$
$$\cdot (\sin(y) + 0.1\sin(20y) + \cos(100y)), \quad \text{(2D Allen-Cahn)} \tag{22}$$
$$u(x, y) = 5x\cos(100x)\sin(y) + x^3 - y^3. \quad \text{(2D Poisson)} \tag{23}$$

For the NN of all the methods, we used three hidden layers with 100 neurons per layer and tanh activation. We used the same settings as in Sec 6.1 to ran all the methods. For RFF-PINN and FourierPINN, we set K = 200. The solution error (over five runs) is reported in Fig. 4. Note that with all the single-scale settings, RFF-PINN failed and gave large relative errors, and we did not report their results in the figure. Obviously, the 2D cases are much more challenging for every method. In

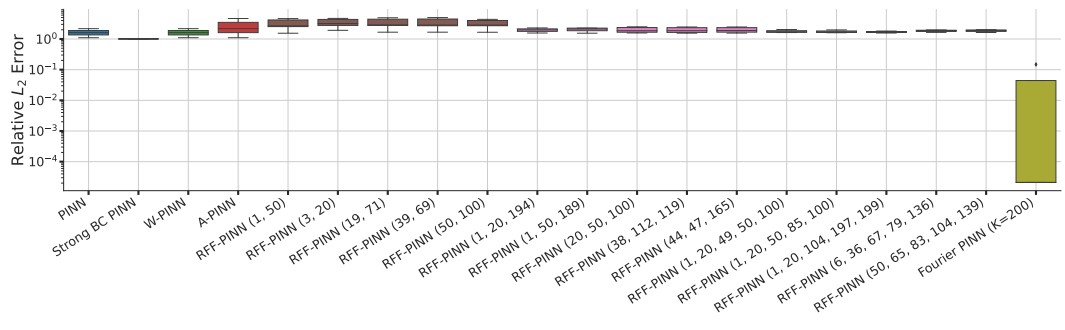

(a) 2D Poisson with $u(x, y) = \sin(100x)\sin(100y)$

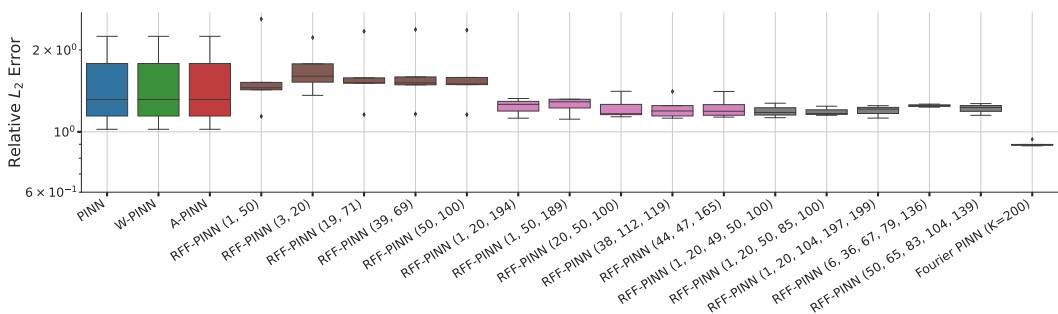

(b) 2D Allen-Cahn with $u(x, y) = (\sin(x) + 0.1\sin(20x) + \cos(100x)) \cdot (\sin(y) + 0.1\sin(20y) + \cos(100y))$

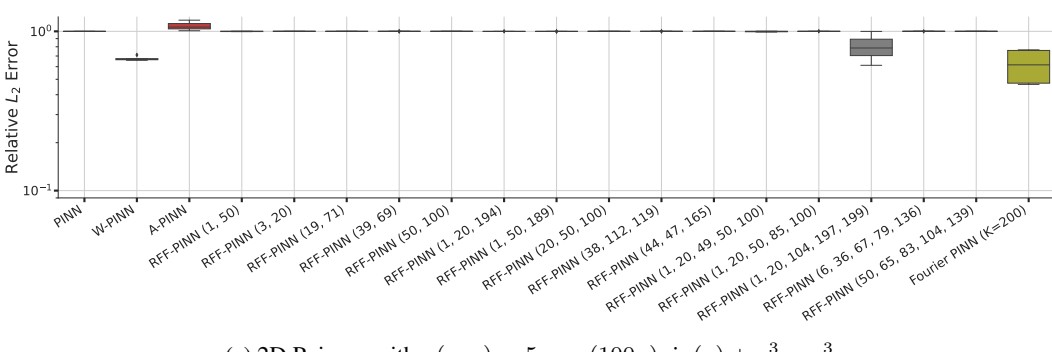

(c) 2D Poisson with $u(x, y) = 5x\cos(100x)\sin(y) + x^3 - y^3$

Figure 4: Solving 2D equations.

particular, for the hybrid case (23), RFF-PINN nearly failed with every choice of the number and scale set. FourierPINN consistently outperforms all the competing methods with all the settings, in most cases by a large margin. Together these results have confirmed the advantage of FourierPINN.

## 7   Conclusion

We have presented FourierPINN, a novel PINN extension to capture high-frequency and multi-scale solution information. Our adaptive basis learning and selection algorithm can automatically identify target frequencies and truncate useless ones, without the need for tuning the number and sales. The performance on a set of benchmark equations is encouraging. Currently, we assume the spacing (granularity) of the candidate frequencies is fine-grained enough, which can be overly optimistic. In the future, we plan to develop adaptive spacing approaches to further capture the granularity.

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

# A Appendix

## 1D Allen-Cahn

We report the relative $L_2$ error for solving 1D Allen-Cahn equation with the Dirichlet boundary condition (see (7)). The exact solution is $u(x) = \sin(100x)$. As shown in Fig. 5, FourierPINN consistantly outperforms all the competing methods.

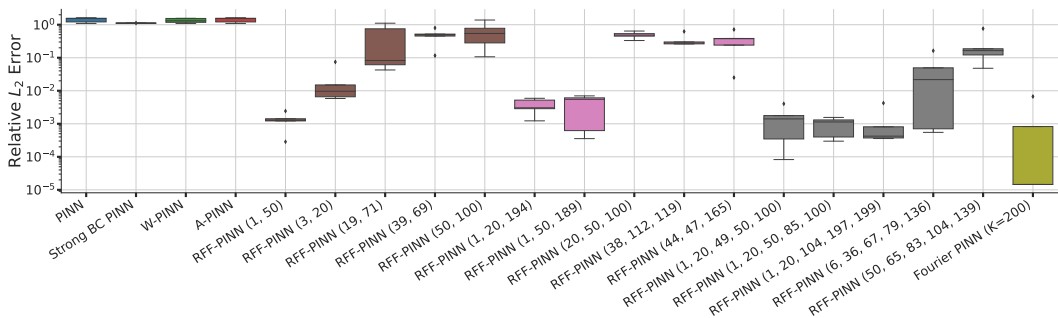

Figure 5: 1D Allen-Cahn equation with true solution $u(x) = \sin(100x)$

