# OpenReview forum: "Fourier PINNs: From Strong Boundary Conditions to Adaptive Fourier Bases"
_ICLR.cc/2023/Conference — Submitted to ICLR 2023_

### Official Review · Reviewer_gWnw · 2022-10-17

**Confidence:** 3
**Correctness:** 3
**Technical Novelty And Significance:** 2
**Empirical Novelty And Significance:** 2
**Recommendation:** 3

**Clarity, Quality, Novelty And Reproducibility:**

The paper is written clearly, the rather weak experimental section makes it hard to judge how significant and how novel the method is. I strongly encourage the authors to improve on this sections. For reproducibility enough information should be provided in the paper.

**Strength And Weaknesses:**

Strengths:
The paper identifies one specific problem of PINNs (capturing high-frequency information) via Fourier analysis on strong BC PINNs, which is quite neat.

Weaknesses:
All the experiments are rather toy-like, and thus it is hard to judge how well the proposed method works. I would like to see at least one temporal PDE, and also some standard PDE problems to test if the approach is still valid there are harms the PINN performance on any other problem besides the toy problems. Which f(x) is needed to get the true solutions outlined in the paper?
PINNs are known for having problems with high-frequency information, there is tons of work on that. Would it be possible to put that into better perspective, both method-wise and comparison-wise? I doubt that RFF PINNs are the only competitors here?
I am missing details on the training behavior of the FourierPINNs, again hard to judge if notorious problems in PINN training can be alleviated by this approach or if they get worse.
One claim of the paper is that FourierPINNs achieve a “faster and more accurate learning of coefficients of higher frequencies”, yet there is nothing in the experiment section which supports the “faster” claim.
Again experiments, I would have liked to see experiments for multiple inputs.


**Summary Of The Paper:**

The paper presents FourierPINN, an extension of PINNs to capture high-frequency information. This is achieved by conducting a Fourier analysis of strong boundary condition PINNs where a specific boundary condition is used. The result is that multiplying the neural network with the boundary function should enable a faster and more accurate learning of the coefficients. The findings are tested on small synthetic datasets.

**Summary Of The Review:**

The paper discusses an interesting extension to PINNs. However, it is hard to judge on the experiments how significant the new method is.

---

### Official Review · Reviewer_9ZeR · 2022-10-24

**Confidence:** 4
**Clarity, Quality, Novelty And Reproducibility:** The topic is interesting, but this wo…
**Correctness:** 3
**Technical Novelty And Significance:** 2
**Empirical Novelty And Significance:** 2
**Recommendation:** 3

**Strength And Weaknesses:**

Both strong boundary condition PINNs and Fourier PINNs achieve good errors in numerical experiments and provide new ideas for solving PDEs with high-frequency and multifrequency
solutions. While the author's approaches are interesting, there are some weaknesses should be pointed out.
1. The main idea of strong boundary condition PINNs is to construct the solution with
$$
\widetilde{u}_{\theta}(x)=B(x)\mathcal{N}_{\theta}(x),
$$
where $\mathcal{N}_{\theta}(x)$ is a neural network parameterized by $\theta$, and $B|_{\partial\Omega}=0$. This approach use be used in penalty-free neural network (PFNN), see Shen and Yang 2021.
2. Furthermore, PFNN extended this approach to complex geometries and nonhomogeneous Dirichlet boundary conditions.
 As we all known, $\{\cos(\pi nx)\}_{n=0}^{\infty}$ (or $\{1\}\cup\{\sin(\pi nx)\}_{n=1}^{\infty}$) is an orthogonal basis of $L^{2}(0,1)$, which means that any function in $L^{2}(0,1)$ can be written as a liner combination of $\{\cos(\pi nx)\}_{n=0}^{\infty}$ (or $\{1\}\cup\{\sin(\pi nx)\}_{n=1}^{\infty}$). The authors should interpret the motivation of using the NN basis in (10) or (11).
3.  Removing the linear combination of NN basis in (10) or (11) yields the spectral methods. In fact, we only need to solve a least square problem in spectral methods, but need to solve non-convex optimization problems many times in Fourier PINNs. I recommend the authors to explain the advantages of Fourier PINNs comparing with the spectral methods.

Ref:
Hailong Sheng and Chao Yang. Pfnn: A penalty-free neural network method
for solving a class of second-order boundary-value problems on complex ge-
ometries. Journal of Computational Physics, 428:110085, 2021.

**Summary Of The Paper:**

Since PINNs often struggle to learn high-frequency and multi-scale target solutions,
authors investigated a strong boundary condition (BC) version of PINNs. By modeling
the solution as an neural network plus the linear combination of a set of dense
Fourier bases, authors developed Fourier PINNs. Finally, authors evaluated Fourier
PINNs in several benchmark PDEs with high-frequency and multi-frequency solutions
and solutions that couple high-frequency components with plain functions. In
all the cases, Fourier PINNs consistently achieve good solution errors.

**Summary Of The Review:**

In summary, although the work is interesting, strong boundary condition does not have enough novelty and the design of Fourier PINNs still have lots of open problems to be solved. There exists plenty of room to improve the quality of the paper at this point, and as such the manuscript should not be published in the present form.

---

### Official Review · Reviewer_nwfE · 2022-10-25

**Confidence:** 3
**Clarity, Quality, Novelty And Reproducibility:** The paper is well written. I wonder w…
**Correctness:** 3
**Technical Novelty And Significance:** 3
**Empirical Novelty And Significance:** 3
**Recommendation:** 5

**Strength And Weaknesses:**

Strength:
- The manuscript compared their results against a lot of baseline methods, which effectively illustrated their performance.

Weaknesses:
- I wonder how this proposed method compares to [1]. In my opinion, the proposed method and [1] both pruned a span of Fourier features but were trained differently by supervised loss and physics-inspired loss.
- The results are very quantitative. It will be interesting to show some visualization of learned PDEs to capture a qualitative sense.
- I would expect more ablation studies of the choices in the proposed methods: for example, how does the number of iterations affect the results?

[1] Li, Zongyi, et al. "Fourier neural operator for parametric partial differential equations." arXiv preprint arXiv:2010.08895 (2020).

**Summary Of The Paper:**

This manuscript proposes Fourier PINNs, a variant of PINN that exactly satisfies a simple boundary condition. In addition to the reinforcement of boundary conditions, this manuscript also did a Fourier analysis of the proposed neural architecture and proposed an efficient implementation of Fourier PINNs that has a better speed and accuracy than previous methods.

**Summary Of The Review:**

The manuscript proposed an effective variant of PINNs which can reinforce simple boundary conditions exactly. I would like to see some explanations and experiments added. I am open to raising the score based on the response.

---

### Decision · Program_Chairs · 2023-01-20

**Decision:**

Reject

**Justification For Why Not Higher Score:**

See above

**Justification For Why Not Lower Score:**

N/A

**Metareview: Summary, Strengths And Weaknesses:**

All reviewers raised several (sometimes severe) points of criticism, and all reviewers voted for rejection. There was no rebuttal, and I also see serious problems with this paper. Although the general idea of FourierPINNs might be interesting and certainly has some potential, this paper just doesn't seem to be ready for publication. Therefore I vote for rejection.